# Synergizing Multi-Plasticizers for a Starch-Based Edible Film

**DOI:** 10.3390/foods11203254

**Published:** 2022-10-18

**Authors:** Jun Fu, Mahafooj Alee, Mao Yang, Hongsheng Liu, Yanan Li, Zhongxian Li, Long Yu

**Affiliations:** 1Institute of Chemistry, Henan Academy of Sciences, Zhengzhou 450002, China; 2School of Materials Science and Engineering, Zhengzhou University, Zhengzhou 450001, China; 3School of Food Science and Engineering, South China University of Technology, Guangzhou 510640, China

**Keywords:** starch film, multi-plasticizer, signalization, packaging, edible

## Abstract

Synergized multi-plasticizers for a starch-based edible film were developed for food packaging. The most popular edible plasticizers, water, glycerol, and sorbitol were used as modal materials to demonstrate the synergized function of multi-plasticizers. The efficiency, stability, and compatibility of each plasticizer, as well as their synergized functions were investigated based on the characterizations of tensile properties after storing under different humidity conditions and for different times. The relationship between the microstructures of the plasticizers and their performances was studied and established. The results showed that water is an efficient plasticizer but is not stable, which results in it becoming brittle under lower humidity conditions; glycerol has a stronger moisture-retaining and absorption capability, which results in a weaker tensile strength under higher humidity conditions; and sorbitol is an efficient and stable plasticizer but needs to work with water, and its function can be synthesized by mixing it with water and glycerol.

## 1. Introduction

Many natural polymers, such as protein, starch, and chitosan have not only attracted considerable attention as eco-friendly materials to replace conventional petrol-based plastic for packaging but have also become ubiquitous as edible packaging films or coatings to protect foods and extend their shelf-life [1,2,3,4,5]. Starch has been shown to be a very promising raw material due to the fact it is biodegradable and edible. Starch is also a kind of renewable and easily obtained resource. The edible films or coatings made from starch have been developed and widely used in food and medicine industries [6,7,8,9,10]. For example, they have been applied in various food (sweet) wrappers and capsules for medicine [11,12,13,14]. Similar to other materials, improvement of the performances and reducing cost are the two strategic aims of starch-based materials. Hydroxypropyl-modified cornstarch (HPCS) has good mechanical and processing characteristics [8,9,10]. A well-recognized weaknesses for starch-based films currently used is the instability of mechanical performances. Under dry conditions the starch film becomes brittle, but in high humility conditions it becomes very soft and loses mechanical strength mainly due to the instability of plasticizers [7,15,16,17] and retrogradation [18].

The International Union of Pure and Applied Chemistry (IUPAC) gave the definition of plasticizers in 1951: it is a substance incorporated in a material matrix to increase its flexibility, workability, or distensibility. The basic requirement for a plasticizer is to be miscible or compatible with the plasticized polymer. Incompatible substances will exudate from the matrix and will result in poor physical properties [19]. All the plasticizers used for starch have similar chemical structures, containing hydroxyl groups, which are compatible with starch that also contain large number of hydroxyl groups. The well accepted three theories for plasticization are lubricity theory, gel theory, and free volume theory [20,21,22]. The ideal plasticizer should meet some basic requirements, such as being compatible and having a low volatility. When the materials are used for food packaging or edible materials, the plasticizer should be nontoxic, odorless, and tasteless [23,24,25]. Water is the most popular and widely used plasticizer for starch since it has been added into starch during various food processes for thousands of years. Water actually acts as both a plasticizer and gelatinization agent [26] in starch-based materials. However, the instability of water makes the starch-based materials very brittle after losing water. Glycerol has been widely added into starch-based materials to keep the moisture [8,9,10]. More recently, the chemicals containing multi-hydroxyl groups have also been developed as plasticizers for starch-based materials, such as sorbitol [27] and xylose, etc. [28]. They are all food ingredients.

The efficiency of a plasticizer containing hydroxyl groups strongly depends on the environment, especially on the humidity conditions. This work aimed to study and develop starch-based edible films used for application in food packaging. The films used for food packing need to meet different humidity conditions. For example, seasonings bags in fast noodles are kept with the noodles constantly under very dry conditions (RH < 5%) while some fruits or candy could be packed in higher humidity conditions. In this work, the most popular edible plasticizers, water, glycerol, and sorbitol, were used as modal materials and various mixtures were studied to explore their plasticizing efficiency and mechanism since glycerol is a liquid while sorbitol is a solid. The efficiency, stability, and compatibility of each plasticizer, as well as their synergized functions were investigated based on the characterizations of the tensile properties after storage under different humidity conditions and for different times. The relationship between the microstructures of the plasticizers and the performances was established. The signalization of different plasticizers were investigated and used as guidelines for developing starch-based edible films.

## 2. Materials and Methods

### 2.1. Materials

All the materials used in this work are commercially available in the market. Hydroxypropyl cornstarch (HPCS) (DS 0.4%, moisture content 13 wt%, Mw 130,000) was supplied by Hengrui Starch Company, Luohe, China since it has good mechanical and processing characteristics [8,9,10]. Tianjin Kemeou (A chemical reagent company, Nanjing, China) provided sorbitol (99.8% pure). Pure glycerol (99%) was purchased from Sinopharm (A chemical reagent company, Shanghai, China).

### 2.2. Sample Preparation

The casting starch films were fabricated from starch suspensions with 10% concentration (in weight *w*/*w*) prepared in a conical flask, in which 10 g starch (dried based) was mixed with 90g water. On a dry basis, the various plasticizers were applied at ratios in proportion to the starch weight (10%). The sample codes and formulations are listed in Table 1. All the materials were pre-mixed before being heated to 98 °C and kept at that temperature for 1 h while being continually shaken. After vigorously shaking for 45 min, the gelatinized starch suspension with various plasticizers was put onto a polystyrene plate. The films were dried in an oven at 35 °C for 10–12 h to obtain a consistent weight. In order to produce the film with same thickness, the same quantity of suspensions was used to put into in the same-sized dish. The moisture content in the film and its thickness were measured after keeping the samples under 25 °C temperature with 20% RH for 7 days.

### 2.3. Mecahnical Testing

The tensile behaviors of the starch films were measured based on the tensile standards of ASTM D882-12 using a tensile facility (Instron 5565). The crosshead of 5 mm/min^−1^ stretching speed under at 25 °C as used in this work. The tensile bar-shaped specimens were cut from the starch film, and were kept in a constant temperature condition with different humidifies (5, 20, 40, 60, 80%) for 48 h (in a condition box from Lab Com., Qingdao, China) before testing. The data presented are the average results from seven specimens.

### 2.4. Microstructure and Morphology Studied by Scanning Electron Microscope (SEM)

The surface structure and fracture surface of the films were studied by a SEM (Thermo Fisher Scientific Inc., NYSE, TMO, Waltham, MA, USA). The fracture surfaces were prepared from tensile testing. The surface and fracture surface of the samples were fixed on metal stubs with double-sided glue then coated under vacuum using an Eiko Sputter Coater. In order to avoid the risk of damaging the surface, a low voltage of 3 kV was used in this work.

### 2.5. UV Transparency Measurement

The film transparency (%) was measured using a UV spectrophotometer (PerkinElmer, Lambda 1050+). The transparency value was presented based on the average of three measurements at a wavelength of 350nm.

### 2.6. Contact Angle (CA)

The CA of water on the starch films was measured using a Contact Angle System OCA20 (Data physics, Germany) at 25 °C. A total of 4 µL water was dropped on the starch film then the value of CA was immediately recorded. Three different places were evaluated for each sample and the reported data were the averages of the three readings.

### 2.7. Statistical Analysis

All the results are presented by date based on statistical analysis as means ± SD (standard deviation). All the experimental results were statistically analyzed by the software ANOVA (IBM, Armonk, NY, USA). The sample with result *p* < 0.05 was classified as having significant differences based on Duncan’s multiple range testing.

## 3. Results and Discussions

### 3.1. Effect of Individual and Mixed Plasticizers

The effects of individual and mixed plasticizers on tensile properties, contact angle, and transparency of starch-based film under a certain RH (40%) are presented in Table 2. It can be seen that both glycerol and sorbitol could replace partly water as a plasticizer for starch-based materials, which was expected since they also contain hydroxyl groups. Both glycerol and sorbitol decreased both modulus and tensile strength, especially with higher content, but increased elongation. It was noticed that glycerol demonstrated more efficiency as a plasticizer than sorbitol. The mixed plasticizers showed a similar trend: by increasing the glycerol/sorbitol content, both tensile modulus and tensile strength were decreased while elongation was increased.

Both glycerol and sorbitol decreased the contact angle, indicating they a stronger water absorption capability, which was expected. The transparency of starch film was not affected by the additional of both glycerol and sorbitol.

Traditional plasticizers for plastics are normally liquids, while the plasticizers for starch-based materials can be either liquids or solids. However, they must be water soluble since these plasticizers have to be used together with water. Table 3 gives the chemical structures and their characterizations of each plasticizer used. All the plasticizers contain hydroxyl groups similar as starch itself. Both water and glycerol are liquid at room temperature. However, glycerol still must work together with water as a plasticizer since the strong hydroxy bound, even the esterification reaction between starch and glycerol under water free conditions, reduces the movement of the starch chain [28]. Sorbitol has a strong water solubility; thus, it should also be in a liquid state when mixed with water.

Natural starches normally contain about 13% moisture [18,29]. This moisture is generally stable in starch particles in different formats including bound or crystalline water or free water without heating. When heating the starch with additional water, the water molecular will diffuse into starch particles and destroy the starch crystalline structure, which is the so-called gelatinization process. The water then acts as a plasticizer to lubricate the starch polymer chains. However, without water, both glycerol and sorbitol cannot gelatinize starch given their larger molecular size. Both glycerol and sorbitol could act as plasticizers after diffusing into starch chains with water. It has been noticed that both glycerol and sorbitol have higher (>290 °C) boiled temperatures, which allows them to be stable in the starch matrix at room temperature. The synergetic action between glycerol and sorbitol enhanced the plasticizing efficiency.

### 3.2. Fracture Interface

Figure 1 shows SEM images of the fracture surface of starch plasticized by different plasticizers after storing at 40% RH. Based on the knowledge that toughened and plasticized materials show a rough fracture surface [9,28,30], it is seen that during the broking of a material by tensile force, the fracture surface of the material shows some deformations for the material with toughness behaviors. It can be seen that the fracture surface of the starch only plasticized by water had a smooth surface with typical brittle broken marks, while the fracture surfaces of starch plasticized by water with both glycerol and sorbitol, as well as their mixtures, showed some deformation of the starch matrix, indicating the toughness behaviors of the materials. This phenomenon can be used to explain the improvement of toughness by the additional of glycerol and sorbitol, as well as their mixtures. The results corresponded with the tensile properties (see Table 2).

### 3.3. Performances under Different Humility Conditions

Figure 2 presents the effect of relative humidity on the mechanical performance of starch-based films containing different plasticizers. It can be seen that water was unstable under dry (low RH) conditions, in which water was lost (evaporated) resulting in brittle (very low elongation) starch materials. Under very low RH conditions (<20%), the starch sheets became too brittle; thus, it was hard to measure their mechanical properties. On the other hand, glycerol has too strong a capability of water absorption, which meant that the material became too soft under high RH. For example, the modulus of the sample containing glycerol (SWG-4) was only about 1/3 of the one containing only water (SW) under 80% RH. The function of sorbitol is between water and glycerol even though it needs to work together with water. Under very low RH conditions (<20%), the starch sheets containing sorbitol also became very brittle. It was noticed that the multi-plasticizers showed a better plasticizing balance. For example, the sample (St-WGS-4) showed a reasonable balance of mechanical properties under both higher and lower RH. The sample St-WGS-4 still showed some elongation in dry conditions and kept a certain modulus and tensile strength under higher RH conditions.

It has been widely reported and well recognized that the performances of starch films strongly depend on plasticizer content and humidity conditions [9,28]; thus, many research works have focused on reducing their moisture sensitivity [31,32,33]. The results have indicated that some plasticizers are very sensitive to environmental factors, especially humidity, which means that the starch-based materials are also sensitive to environmental factors. For example, given the strong water absorption behavior, glycerol could absorb water under high RH, which results in the weakness of the starch materials. In the review of their applications, a multi-plasticizer system should be studied and developed. This work has provided guidelines and directions.

## 4. Conclusions

The accepted plasticizers used for starch films, water, glycerol, and sorbitol, were used as modal materials to study the plasticizing efficiency and performance under different environmental conditions, as well as to demonstrate the synergized function of multi-plasticizers. Water acts as both a plasticizer and gelatinization additive for starch. Water is a very efficient plasticizer but is not stable, especially under very dry conditions, which results in the brittleness of starch materials under lower humidity conditions. Other plasticizers, including both liquid glycerol and solid sorbitol must work together with water as plasticizers for starch films. Without water, both glycerol and sorbitol cannot gelatinize starch given their larger molecular size. Both glycerol and sorbitol could act as plasticizers after penetrating starch chains with water. It was found that glycerol has a stronger moisture-retaining and absorption capability, which results in a weaker tensile strength under higher humidity conditions; while sorbitol is an efficient and stable plasticizer but needs to work with water, and its function can be synthesized by being mixed with water and glycerol. The fracture surface of the starch only plasticized by water had a smooth surface with typical brittle broken marks, while the fracture surfaces of starch plasticized by water with both glycerol and sorbitol, as well as their mixtures, showed some deformation of the starch matrix, indicating the toughness behavior of the materials. It was noticed that the multi-plasticizers showed a better plasticizing balance of the mechanical properties under both higher and lower RH. The sample containing both water, glycerol, and sorbitol showed some elongation under dry conditions and kept certain modulus and tensile strength under higher RH conditions. The results provide useful guidelines and directions for developing plasticizers for starch-based materials.

## Figures and Tables

**Figure 1 foods-11-03254-f001:**
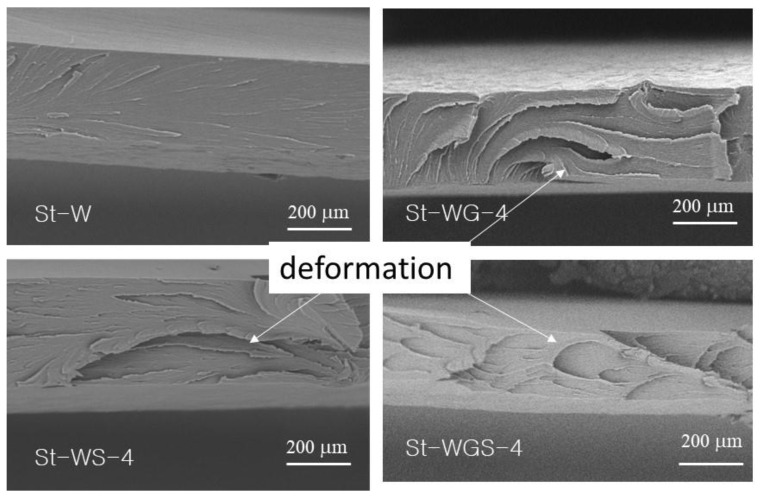
SEM image of the fracture surface of starch-based films containing different plasticizers.

**Figure 2 foods-11-03254-f002:**
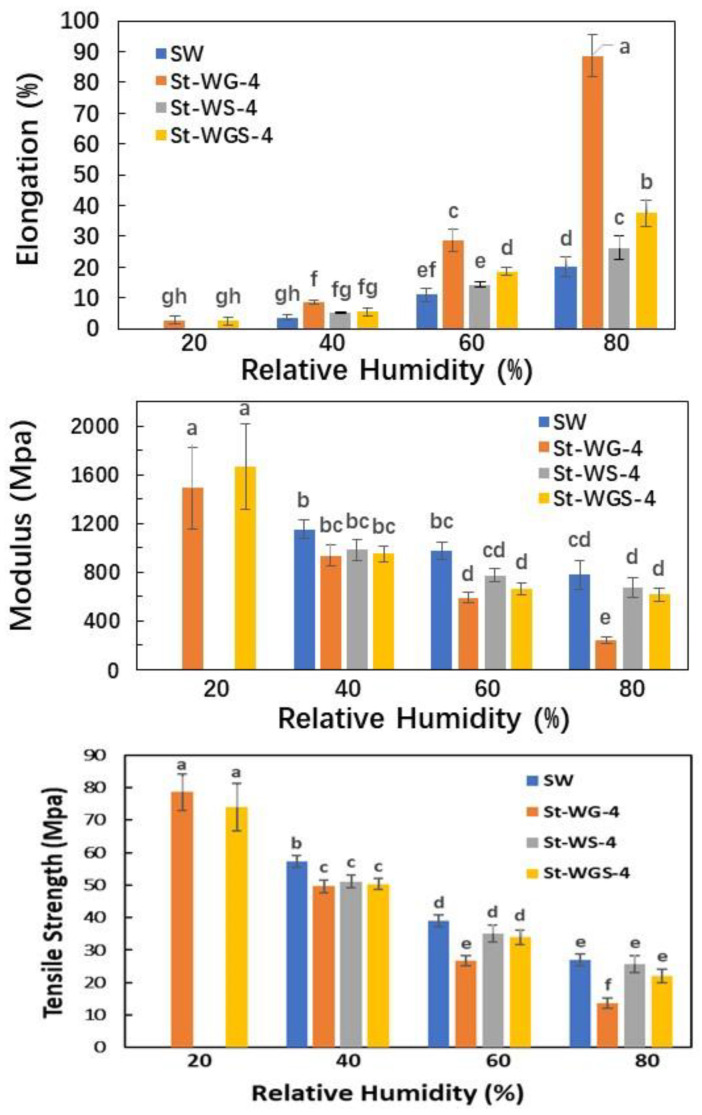
Effect of humidity conditions on the tensile properties of a starch film plasticized with different and multi-plasticizers. The results were statistically analyzed by ANOVA and marked as abcdefgh.

**Table 1 foods-11-03254-t001:** Formulations, sample codes, and the film characterizations.

Sample Code ^1^	Starch (*w*/*w*%)	Water (*w*/*w*%)	Glycerol (*w*/*w*%)	Sorbitol (*w*/*w*%)	Thickness (μm)	Moisture Content (%) ^2^
St-W	10	90	-	-	156 ± 4	12.01
St-WG-1	10	89.5	0.5	-	160 ± 8	12.62
St-WG-2	10	89	1	-	171 ± 9	12.83
St-WG-3	10	88	2	-	173 ± 8	13.73
St-WG-4	10	87	3	-	181 ± 9	13.87
St-WS-1	10	89.5	-	0.5	167 ± 7	12.27
St-WS-2	10	89	-	1	171 ± 6	12.57
St-WS-3	10	88	-	2	177 ± 8	12.91
St-WS-4	10	87	-	3	172 ± 6	13.11
St-WGS-1	10	88	0.25	0.25	152 ± 8	12.37
St-WGS-2	10	85	0.5	0.5	168 ± 6	12.59
St-WGS-3	10	83	1	1	167 ± 7	13.21
St-WGS-4	10	80	1.5	1.5	165 ± 6	13.65

Notice: ^1^ W—water, G—glycerol, S—sorbitol. ^2^ The specimens were kept under RH of 40% at 25 °C for 7 days.

**Table 2 foods-11-03254-t002:** Effect of different plasticizers on mechanical properties, contact angle, and transparency.

Sample Code	Modulus (MPa)	Tensile Str (MPa)	Elongation (%)	Contact Angle (Ɵ)	Transparency (% at 260 nm)
St-W	1152 ± 119 ^ab^	57.3 ± 2.9 ^a^	3.6 ± 0.7 ^cd^	93.0	86.1
St-WG-1	1127 ± 102 ^bcd^	55.5 ± 4.8 ^abc^	4.3 ± 0.7 ^bc^	89.6	86.4
St-WG-2	1029 ± 116 ^cd^	52.2 ± 4.6 ^bc^	4.4 ± 0.9 ^bc^	87.5	84.3
St-WG-3	992 ± 72 ^cde^	51.2 ± 5.2 ^bcd^	6.3 ± 0.8 ^b^	85.7	85.7
St-WG-4	933 ± 86 ^e^	49.4 ± 5.6 ^d^	8.6 ± 0.6 ^a^	83.9	85.3
St-WS-1	1182 ± 112 ^a^	56.8 ± 3.9 ^b^	3.4 ± 2.1 ^d^	91.4	84.8
St-WS-2	1120 ± 106 ^bcd^	54.7 ± 4.8 ^bc^	4.1 ± 1.6 ^bc^	90.8	85.2
St-WS-3	1080 ± 76 ^cd^	52.7 ± 4.8 ^bc^	4.8 ± 1.6 ^bc^	87.4	86.7
St-WS-4	982 ± 84 ^de^	51.1 ± 5.9 ^bcd^	5.3 ± 0.2 ^bc^	87.3	86.5
St-WGS-1	1142 ± 79 ^abc^	55.6 ± 3.2 ^ab^	3.6 ± 1.2 ^cd^	91.5	85.1
St-WGS-2	1067 ± 74 ^cd^	54.5 ± 3.1 ^bc^	4.3 ± 1.5 ^bc^	88.2	85.6
St-WGS-3	998 ± 81 ^cd^	51.9 ± 2.3 ^bc^	4.9 ± 2.1 ^bc^	86.2	85.1
St-WGS-4	950 ± 66 ^e^	50.3 ± 1.7 ^cd^	5.4 ± 1.3 ^bc^	86.9	85.8

All the specimens were kept at room temperature with RH 40% for 4 days. The results were statistically analyzed by ANOVA and marked as abcde.

**Table 3 foods-11-03254-t003:** Chemical structures and their characterizations of various plasticizers used in this work.

Materials	Chemical Structure	Physical State (Dry)	Melting/Boiling Temp (°C)	Viscosity *(Pa)	Solubility (g/100 g Water)
water	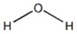	liquid	0/100	1.0	-
glycerol	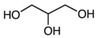	liquid	17.9/290	1.5	miscible
sorbitol	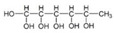	powder	95/295	1.4 *	235

* 10% wt. concentration in water.

## Data Availability

The data presented in this study are available on request from the corresponding author.

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
