# Peer review of "Synergizing Multi-Plasticizers for a Starch-Based Edible Film"

_foods, 2022, doi:10.3390/foods11203254_

Round 1

Reviewer 1 Report

The authors tested the effect of combined plasticizers in starch-based materials. 

The introduction needs more explanation about the kind of material they are using. For example they refer to native starch in the introduction while the characterization has been made on HP-starch that is notably more flexible and less prone to retrogradation. 

10 % starch solution? what is the molecular weight of this starch? what degrees of hydroxypropylation has? The plasticizers were combined at what stage with the solution? Since glycerol and sorbitol are different in terms of hygroscopicity how did you treat them? since the MM are so different did you consider it when mixing it in the solution?

Since it is an HP starch what temperature did you use to solubilize it? This material after the modification gets hydrolyzed to some extent so I wouldn't talk about gelatinized starch since it has been already gelatinized by the company that produced it.

you used ASTM D882-12, please write in the MeM text what are these conditions.

the 2.5 paragraph in MeM says Transparency Measurement - Please correct it in UV transparency Measurement

The manuscript lacks discussion and comparison with old and recent literature. in general, the authors need to improve it both in the discussion and conclusion. 

Author Response

Responses to Reviewer-1

The introduction needs more explanation about the kind of material they are using. For example they refer to native starch in the introduction while the characterization has been made on HP-starch that is notably more flexible and less prone to retrogradation. 

Response: We understand this question and added more introduction.

10 % starch solution? what is the molecular weight of this starch? what degrees of hydroxypropylation has? The plasticizers were combined at what stage with the solution? Since glycerol and sorbitol are different in terms of hygroscopicity how did you treat them? since the MM are so different did you consider it when mixing it in the solution?

Response: We have added this information in the Materials section, and more discussions. Generally, the compatibility between starch and glycerol and sorbitol mainly depends on the hydroxy groups on both materials, which results that the effect of MW has limited.

Since it is an HP starch what temperature did you use to solubilize it? This material after the modification gets hydrolyzed to some extent so I wouldn't talk about gelatinized starch since it has been already gelatinized by the company that produced it.

Response: The HP modified has lower gelatinization temperature than that of raw starch so the 98 C was used to prepare the starch suspension. We have added this information.

you used ASTM D882-12, please write in the MeM text what are these conditions.

Response:

the 2.5 paragraph in MeM says Transparency Measurement - Please correct it in UV transparency Measurement

Response: We agree with this suggestion and corrected it.

The manuscript lacks discussion and comparison with old and recent literature. in general, the authors need to improve it both in the discussion and conclusion. 

Response: We have added more discussions.

Reviewer 2 Report

The work deals with the study of the possibilities to improve some physical properties of starch based materials, used especially for food packaging. For these purpose different plasticizers as water, glycerol and sorbitol were used as modal materials in different mixtures.

The efficiency, stability, and compatibility of each plasticizer, were estimated from the tensile properties after storing the samples under different humidity conditions and different times. The study tries to establish a relationship between the microstructures of the plasticizers and performances of the materials.

The samples were prepared as films following standard protocols, and were conditioned in a temperature-humidity box with different humidifies for 48 h before testing. The following investigations were done: Tensile Testing, Scanning Electron Microscopy (SEM), Transparency Measurement, Contact Angle (CA), followed by Statistical Analysis.

The authors concluded that both glycerol and sorbitol can replace partly the water as plasticizer for starch-based materials, due to theirs hydroxyl groups. The glycerol and sorbitol decreases the modulus and tensile strength, and increases elongation.

In addition the glycerol and sorbitol decreases the contact angle, but they didn’t affect the transparency of the films. 

            1. It is suitable to include in the paper some curves of light absorption.

            2. What is the reason to use the absorption investigations, what information could be obtained from this analysis?

Some explanations concerning the role of water as plasticizer were done.  

            3. More references must be provided.    

SEM investigation of fracture interface shows differences between the samples with different plasticizers.

            4. What is the reason to use the SEM investigations, what information could be obtained from this analysis? Here the discussion is very short.

Practically results concerning the elongation, modulus and tensile strength for films with different plasticizers in function of humidity conditions were obtained.

            As conclusion the authors affirm that water is a very efficient plasticizer but not stable, especially under very dry condition, which results in brittleness of starch materials.

            The glycerol and solid sorbitol must be used together with water as plasticizer for starch films to improve some mechanical properties of the samples. The glycerol has strong moisture retaining and absorption capability.

            It is an experimental work, properly done, with results of practically application, but without a solid theoretical support.

            5. The paragraph ”discussions” must be supported by more references.

Author Response

Reviewer-3

The work deals with the study of the possibilities to improve some physical properties of starch based materials, used especially for food packaging. For these purpose different plasticizers as water, glycerol and sorbitol were used as modal materials in different mixtures.

The efficiency, stability, and compatibility of each plasticizer, were estimated from the tensile properties after storing the samples under different humidity conditions and different times. The study tries to establish a relationship between the microstructures of the plasticizers and performances of the materials.

The samples were prepared as films following standard protocols, and were conditioned in a temperature-humidity box with different humidifies for 48 h before testing. The following investigations were done: Tensile Testing, Scanning Electron Microscopy (SEM), Transparency Measurement, Contact Angle (CA), followed by Statistical Analysis.

The authors concluded that both glycerol and sorbitol can replace partly the water as plasticizer for starch-based materials, due to theirs hydroxyl groups. The glycerol and sorbitol decreases the modulus and tensile strength, and increases elongation.

In addition the glycerol and sorbitol decreases the contact angle, but they didn’t affect the transparency of the films. 

  1. It is suitable to include in the paper some curves of light absorption.

Response: We understand this concern. Since this edible film can only be used as internal packaging (due to moisture sensitive) we have not considered its light absorption issue yet.

  1. What is the reason to use the absorption investigations, what information could be obtained from this analysis?

Response: Since the strong water absorption behavior of some plasticizers (with hydroxyl groups, like glycerol) they could absorb water under high RH, which results in weakness of the starch materials. In the review of application, these kinds of substance are not suitable for applications as plasticizers, or at least the content cannot be higher.

Some explanations concerning the role of water as plasticizer were done.  

  1. More references must be 

Response: We agree with this comment. Since there are too many of them it is hard to select them.

SEM investigation of fracture interface shows differences between the samples with different plasticizers.

  1. What is the reason to use the SEM investigations, what information could be obtained from this analysis?Here the discussion is very short.

Response: The reason of applicating SEM is to study their fracture surface then to establish the relationship between microstructure and performance. Based on the materials knowledge, toughened or plasticized materials have rough fracture surface. We agree with this suggestion and have added more discussions.  

 Practically results concerning the elongation, modulus and tensile strength for films with different plasticizers in function of humidity conditions were obtained.

            As conclusion the authors affirm that water is a very efficient plasticizer but not stable, especially under very dry condition, which results in brittleness of starch materials.

            The glycerol and solid sorbitol must be used together with water as plasticizer for starch films to improve some mechanical properties of the samples. The glycerol has strong moisture retaining and absorption capability.

            It is an experimental work, properly done, with results of practically application, but without a solid theoretical support.

  1. The paragraph ”discussions” must be supported by more references.

Response: 55We agree with this suggestion and added more.

Reviewer 3 Report

The article examine the use of popular plasticizers as an additive to starch based edible film. The authors investigate how the addition of water, glycerol and/or sorbitol affects the mechanical properties, microstructure, morphology and contact angle of the starch film. The authors also analyze the synergistic effect of the plasticizers used.

The methodology is described correctly. The results are clear and well explained. However the article would be more valuable if the obtained results were contrasted with other literature data e.g.

Basiak, E.; Lenart, A.; Debeaufort, F. How Glycerol and Water Contents Affect the Structural and Functional Properties of Starch-Based Edible Films. Polymers 2018, 10, 412. https://doi.org/10.3390/polym10040412

Laohakunjit, N., Noomhorm, A. Effect of Plasticizers on Mechanical and BarrierProperties of Rice Starch Film Starch/Stärke56(2004)  348–356

Paluch, M., Ostrowska, J., TyÅ„ski, P. et al. Structural and Thermal Properties of Starch Plasticized with Glycerol/Urea Mixture. J Polym Environ 30, 728–740 (2022). https://doi.org/10.1007/s10924-021-02235-x

Moreover the authors do not explain why they chose to use hydroxypropylated corn starch in their study.

It is difficult for me to pinpoint the exact place of the correction because the manuscript lacks line numbering.

Page 2 last line in “Sample preparation” the abbreviation RH should be explained

Table 1. Authors use in table header as one of plasticizer “Xylose”. Should not it be sorbitol?

Page 3. Section 2.6 – line 3 - what is the meaning of CV abbreviation?

Based on the high quality of the manuscript I suggest it to be accepted after minor revision.

Author Response

Responses to Reviewer-2

The article examine the use of popular plasticizers as an additive to starch based edible film. The authors investigate how the addition of water, glycerol and/or sorbitol affects the mechanical properties, microstructure, morphology and contact angle of the starch film. The authors also analyze the synergistic effect of the plasticizers used.

The methodology is described correctly. The results are clear and well explained. However the article would be more valuable if the obtained results were contrasted with other literature data e.g.

Basiak, E.; Lenart, A.; Debeaufort, F. How Glycerol and Water Contents Affect the Structural and Functional Properties of Starch-Based Edible Films. Polymers 2018, 10, 412. https://doi.org/10.3390/polym10040412

Laohakunjit, N., Noomhorm, A. Effect of Plasticizers on Mechanical and BarrierProperties of Rice Starch Film Starch/Stärke56(2004)  348–356

Paluch, M., Ostrowska, J., TyÅ„ski, P. et al. Structural and Thermal Properties of Starch Plasticized with Glycerol/Urea Mixture. J Polym Environ 30, 728–740 (2022). https://doi.org/10.1007/s10924-021-02235-x

Response: We have added these references

Moreover the authors do not explain why they chose to use hydroxypropylated corn starch in their study.

Response: We have added these information

It is difficult for me to pinpoint the exact place of the correction because the manuscript lacks line numbering.

Response: This is the format of this journal.

Page 2 last line in “Sample preparation” the abbreviation RH should be explained

Response: We have added these information.

Table 1. Authors use in table header as one of plasticizer “Xylose”. Should not it be sorbitol?

Response: We have corrected it.

Page 3. Section 2.6 – line 3 - what is the meaning of CV abbreviation?

Response: We have corrected it.

Based on the high quality of the manuscript I suggest it to be accepted after minor revision.

Round 2

Reviewer 1 Report

The authors partially answered the questions, but the manuscript still lacks discussion and comparison with the literature. It could work as a short communication 

Author Response

We have added more discussions.